# Response of Apple Flesh to Compression under the Quasi-Static and Impact Loading Conditions

**DOI:** 10.3390/ma15217743

**Published:** 2022-11-03

**Authors:** Zbigniew Stropek, Krzysztof Gołacki

**Affiliations:** Department of Mechanical Engineering and Automation, University of Life Sciences in Lublin, Głęboka 28, 20-612 Lublin, Poland

**Keywords:** apple, toughness, firmness, elasticity modulus, failure stress and strain

## Abstract

Compression tests of cylindrical samples were carried out using two ‘Rubin’ and ‘Red Jonaprince’ apple cultivars with flesh firmness differing in a statistically significant way. The tests were conducted under both the quasi-static and impact loading conditions, which required the use of two test stands. For this purpose, an impact measuring stand was designed and built. The tests proved that the firmness of the apple flesh influenced the mechanical response under both the quasi-static and impact loading conditions. The elasticity modulus had much higher values under the impact than quasi-static loading conditions. This indicates that the stiffness of the apple tissue was larger during the impact than at the low-speed compression. Different failure mechanisms of cylindrical apple flesh samples were found depending on the loading conditions. Under the quasi-static loading conditions, the apples of both cultivars were damaged at the same strain value. However, during the impact, apples were apt to damage at a constant stress value regardless of the firmness of the tested cultivar. The toughness of the apple tissue depended on firmness and was larger under the quasi-static loading conditions for the apples with larger firmness. However, under the impact loading conditions, the toughness was greater for the apples with smaller firmness.

## 1. Introduction

Knowledge of mechanical damage occurring in agricultural products is essential to reduce losses and determine the quality of fruit and vegetables that is downgraded during transport, storage and handling [1,2,3,4]. The complex structure of plant materials causes mechanical properties to be affected by many factors such as: temperature, turgor, shape and size, fruit orientation relative to the loading direction, ripeness, and microstructure [5,6,7]. An additional difficulty is the heterogeneity of the material, the properties of which change over time.

The behaviour of plant material under load is most often described by the force–deformation curves in the compression, tension or puncture tests [8,9,10,11]. From these curves, the following mechanical parameters can be determined: modulus of elasticity, toughness, bioyield strength, failure stress and failure strain.

An important mechanical parameter of agricultural products is the elasticity modulus [12,13,14]. It is defined as the ratio of stress to strain in the elastic region and describes the tissue stiffness of the fruit flesh. The elasticity modulus allows one to estimate and predict changes in the shape of agricultural products during compression and, subsequently, to determine forces leading to excessive deformation resulting in material failure. This can be useful for improving food and agricultural machinery. As the modulus of elasticity is an important parameter for determining the bruising of apples, this is used in designing sorting lines, reducing waste or improving fruit quality. For example, in the design of apple packing lines, knowledge of many parameters such as allowable drop height, firmness and tissue stiffness (modulus of elasticity) that affect apple bruising is required [15].

Toughness is the work required to damage a material per unit volume. This quantity is determined by calculating the area under the stress–strain or force–deformation curve. While using a force–deformation curve, the dimensions of the specimen must be taken into account. In many studies, this area under the curve was simplified to the triangle area formula 1/2*σ_f_⋅ε_f_* [16].

Mohsenin [17] defined the biological yield strength as the point on the force–deformation or stress–strain curve where there is a decrease or no change in force as the strain increases. The biological yield point defines the beginning of damage in the cellular structure of a material. Knowledge of this point allows one to determine the failure stress and strain under the given loading conditions. The generation of different states of stress and strain in the fruit or its sample allows the determination of the failure criterion of the plant tissue. Several ways of establishing a criterion for plant tissue failure are presented in the literature. It was commonly believed that the causes of damage in the fruit tissue were due to tangential stresses. When the cylindrical specimens were compressed, the destruction usually occurred along a plane inclined at 45° to the force direction. This type of failure was reported by many researchers [18,19,20]. However, under certain conditions, specimens fail along a plane perpendicular to the force direction. This indicates that failure can be caused by either normal stress [21] or normal strain [22,23]. The normal critical strain hypothesis was confirmed by Chen and Sun [24], with the reservation that it is not valid over a wide range of strain rates.

Damage occurs in each stage of postharvest operations. Hence, fruit is subjected to dynamic forces during harvesting (transport and placing in bins in the field), followed by sorting, grading and other handling operations [25,26,27]. However, static loads take place during the storage. Therefore, studies on the fruit strength should include both types of these loads [28,29]. Conducting experiments under different loading conditions (different deformation velocities) makes it possible to determine the criterion of apple tissue flesh damage during the compression of cylindrical samples under the quasi-static and impact loading conditions.

Therefore, the aim of the study was:−to determine the biological yield point of apple flesh with different firmness by determining the failure stress *σ_f_* and the failure strain *ε_f_*;−to show the different response of apple flesh under quasi-static and impact loading conditions;−to determine the stiffness and strength of the apple flesh tissue calculating the elastic modulus *E* and the toughness *T*.

## 2. Materials and Methods

### 2.1. Material

The cylindrical samples of apples of the ‘Rubin’ and ‘Red Jonaprince’ cultivars with a height and diameter of 20 mm were tested. Three cylinders were cut from each apple to carry out compression tests at three different deformation velocities for the tests both under the quasi-static and impact loading conditions. This resulted in a smaller scatter of measurement results and a more accurate determination of the response of the flesh of the same apple at different deformation velocities for a given type of loading. A punching die with an internal diameter of 20 mm was used to cut the samples. The samples were cut parallel to the vertical axis of the apple passing through the calyx and stem. The cut out cylinder was then placed in a sleeve with the 20 mm internal diameter and height to achieve the same size.

### 2.2. Compression Tests

The specimens were compressed along the axis until their failure. This was manifested by a decrease in the force response when the maximum value was reached. For both types of tests, the specimens were compressed between two smooth metal plates. The use of smooth surfaces as opposed to rough surfaces did not confine the lateral displacement of the specimen, and hence the stress state in the specimen was more like unidirectional [30,31]. For the experiment under the impact loading conditions, the sampling rate of 153.6 kHz was used to obtain a great accuracy in recording the force response. Under the quasi-static loading conditions, the sampling frequency was 100 Hz. The compression tests under the quasi-static loading conditions were carried out using the TA.HD plus a texture analyser for the three velocities: 0.0001 m⋅s^−1^, 0.001 m⋅s^−1^ and 0.01 m⋅s^−1^. Measurements under the impact loading conditions were performed on an impact measuring stand [32] for the velocities: 0.4 m⋅s^−1^, 0.7 m⋅s^−1^ and 1 m⋅s^−1^. The tests were carried out at room temperature. For each deformation velocity, 10 repetitions were made. As the tests were conducted for three deformation velocities and under two loading conditions, the total of 120 compression tests were carried out.

### 2.3. Firmness Measurements

In order to extend the research, apple cultivars whose firmness differed in a statistically significant way were selected. The firmness was measured with the Magness-Taylor penetrometer. This consisted in reading the force required to press the 11 mm diameter plunger into the apple flesh to the depth of 8 mm. The test was carried out after removing the peel in the central part of the fruit. The penetrometer was placed in the universal stand to ensure linearity of movement. The effect of temperature was eliminated by placing the apples at room temperature for 12 h before testing. Ten apples of each cultivar were used for testing. Three replicates were made for each apple to eliminate differences in the flesh firmness within a single fruit (red and green sides). The accuracy of the firmness measurement was 1 N. The soft cultivar ‘Rubin’ and the hard cultivar ‘Red Jonaprince’ were selected for the study.

### 2.4. Measuring Stands

The impact measuring stand (Figure 1) for the cylindrical specimens consisted of a pendulum with a rigid arm ending in a cylindrical hammer in which a piezoelectric force sensor was placed. The mass of the hammer was chosen so that the kinetic energy was much greater than the deformation energy of the specimen. Thus, it could be assumed that the sample was deformed at the constant velocity. The length of the pendulum arm was 940 mm and the displacement of the hammer during the contact with the specimen was a few millimeters. Thus, the deformation of the specimen could be assumed to be along the straight line. The anvil in the form of a cylinder was fixed to the concrete vertical wall. The cylindrical specimen was fixed with the technical vaseline to the vertical plate screwed into the anvil. The WMU45SK non-contact angle sensor was mounted to the pendulum axis to measure the angle of deflection of the pendulum arm from the vertical and thus to determine the drop height and deformation velocity. At the impact velocities of 0.4 m⋅s^−1^, 0.7 m⋅s^−1^, 1 m⋅s^−1^ and with the pendulum arm length of 940 mm, this corresponded to the pendulum arm deflection angle values of 7.6°, 13.2° and 18.9°, respectively.

The force measurement under the impact loading conditions was made using the Endevco piezoelectric force sensor, model 2311-100 with the sensitivity of 24.23 mV/N and the measuring range of ±220 N (Endevco, Sunnyvale, CA, USA). The microprocessor-based recorder with the sensor was used to transfer data to a computer. The sampling frequency of the microprocessor recorder was 153.6 kHz. Under the quasi-static loading conditions, the tests were carried out using the two-column TA.HD plus texture analyser (Figure 2) of the Stable Micro Systems company equipped with the 300 N measuring head (Stable Micro System, Goldaming, UK). The sampling frequency during testing was 100 Hz. The measurement was triggered when the force response exceeded 0.1 N.

### 2.5. Way of Making Measurements

The tests consisted in recording the force response and displacement during the compression and impact of a cylindrical specimen. The maximum force response and the corresponding displacement were determined from the force–displacement curve. This allowed the failure stress *σ_f_* and the failure strain *ε_f_* to be determined from the equation:(1)$$\sigma_{f}=\frac{F_{\max}}{A}$$
(2)$$\varepsilon_{f}=\frac{\Delta l}{l}$$
where *F*_max_—the maximum force response (N); Δ*l*—the deformation corresponding to the maximum force response (mm); *A*—the cross-section area of the specimen (mm^2^); *l*—the initial length of the specimen (mm).
(3)$$A=\frac{\pi \cdot d^{2}}{4}$$
(4)$$\Delta l=l-l_{\max}$$
where *d*—the specimen diameter (mm); *l*_max_—the specimen length corresponding to the maximum force response (mm).

The elasticity modulus *E* was determined as the slope of a linear section of the stress–strain curve. Two points corresponding to the values of 0.8 *σ_f_* and 0.6 *σ_f_* were used to determine the slope. Then, the corresponding strains were read for the above stresses. This made it possible to determine the tangent of the slope angle of the linear section of the stress–strain curve.

The tissue toughness *T* was determined as the area under the stress–strain curve calculated from the beginning of the deformation up to the maximum force response. Hence, the tissue toughness *T* was calculated from the equation:(5)$$T=\frac{\int_{0}^{\Delta l}F\left(l\right)\cdot dl}{A\cdot l}$$

## 3. Results

### 3.1. Firmness

Figure 3 shows the apple flesh firmness for two ‘Rubin’ and ‘Red Jonaprince’ cultivars. Their mean values were 56 and 75 N, respectively, and differed in a statistically significant way.

### 3.2. Elasticity Modulus

As follows from Figure 4, there was a significant difference in the value of the elasticity modulus between the quasi-static and impact loading conditions. Under the quasi-static loading conditions, the modulus of elasticity was in the range of (3–5) MPa depending on the apple cultivar. However, during the impact, the elastic modulus was close to 9 MPa for both apple cultivars. For the first three deformation velocities (0.0001–0.01 m⋅s^−1^), the elasticity modulus increased with the increasing deformation velocity. The statistical significance was found for the differences between the mean values of the elasticity modulus and the deformation velocity under the quasi-static loading conditions. For the ‘Rubin’ cultivar with smaller firmness, the elasticity modulus varied from 2.7 MPa to 3.4 MPa, and for the ‘Jonaprince’ cultivar with larger firmness, it varied from 3.6 MPa to 4.6 MPa. Under the impact loading conditions, the elasticity modulus maintained a constant value of about 9 MPa regardless of the apple cultivar firmness.

### 3.3. Toughness

Toughness varied in a similar way regardless of the tested apple cultivar. In both cultivars, a sudden decrease in toughness between the quasi-static and impact loading conditions was observed. Under the quasi-static loading conditions, the toughness of the ‘Red Jonaprince’ cultivar was larger than that of the ‘Rubin’ cultivar. However, under the impact loading conditions, the study showed that the ‘Rubin’ cultivar had larger toughness than the ‘Red Jonaprince’ one. The firmness of the apple flesh had an effect on the toughness of the tested cultivar. At each deformation velocity under both the quasi-static and impact loading conditions, statistically significant differences were found between the average values of toughness for the two apple varieties. The toughness for the ‘Rubin’ cultivar had a constant value of about 22 kJ⋅m^−3^ and 14 kJ⋅m^−3^ under the quasi-static and impact loading conditions, respectively. However, for the ‘Red Jonaprince’ cultivar, the toughness was 37 kJ⋅m^−3^ and 10 kJ⋅m^−3^ for the quasi-static and impact loading conditions, respectively (Figure 5).

Thus, the apples with larger firmness (‘Red Jonaprince’) had larger toughness at small deformation velocities and smaller toughness during the impact than the apples with smaller firmness (‘Rubin’). This was due to the larger failure stress of the ‘Red Jonaprince’ cultivar under the quasi-static loading conditions, and the smaller failure strain under the impact loading conditions compared to the same quantities of the ‘Rubin’ cultivar.

### 3.4. Failure Stress and Strain

For the failure strain, a sudden decrease was observed depending on the type of the applied loading. Under the quasi-static loading conditions, the failure strain had the constant value regardless of the tested apple cultivar. For the ‘Jonaprince’ cultivar with greater firmness, the failure strain was 0.15, and for the ‘Rubin’ cultivar with smaller firmness, it was 0.14. Under the impact loading conditions, the failure strain was found to be constant with the increasing deformation velocity for both apple cultivars. In the range of the deformation velocity 0.4–1 m⋅s^−1^, the failure strain for the hard cultivar ‘Red Jonaprince’ and the soft cultivar ‘Rubin’ was 0.05 and 0.07, respectively. While the values of the failure strain for the studied apple cultivars under the quasi-static loading conditions did not differ between each other, a statistically significant difference between the mean values of the failure strain for both cultivars was found under the impact.

The relationship between the failure stress and the deformation velocity depended on the apple flesh firmness of the tested cultivars. For the ‘Red Jonaprince’ with a larger firmness, the failure stress under the quasi-static loading conditions reached the constant value of 0.47 MPa. However, a sudden drop to 0.37 MPa was observed under the impact loading conditions. For the ‘Rubin’ cultivar with smaller firmness, the failure stress increased with the increasing deformation velocity, reaching the value of approximately 0.4 MPa for the three largest deformation velocities. Under the quasi-static loading conditions, the mean values of the failure stress between the cultivars differed in a statistically significant way. The situation was different under the impact loading conditions, where no statistically significant differences were found between the mean values of failure stress for the two cultivars.

As follows from Figure 6 and Figure 7, the apple flesh under the quasi-static loading conditions (the first three deformation velocities) failed at the constant strain value regardless of the firmness of the tested cultivar. However, under the impact loading conditions, damage occurred at the constant stress value regardless of the apple cultivar firmness. This means that, depending on the applied deformation velocity (quasi-static or impact loading conditions) regardless of firmness, the apple tissue was subject to different failure criteria.

## 4. Discussion

The studies on the fruit strength proved that they behave differently depending on the specific loading. The attempts were made to explain this at the cellular level by giving different mechanisms of tissue damage. Pitt [33] distinguished two modes of tissue structure damage: failure of the cell walls and failure of the intercellular bonds. The dominance of one of those failure modes results from the relative strength of one of two structural elements. The failure mode was characteristic of a given species and even of fruit or vegetable cultivar and could change depending on the ripeness extent. The failure by the cell debonding was typical of the potato tissue. The failure by the cell wall rupture could be observed in the fresh apple tissue. The other investigations showed that there are two different physical phenomena dependent on the strain rate which cause apple damage [34,35]. The Preston’s research results [36] proved that the cellulose microfibrils in the cell walls can undergo damage by slipping or breaking primary bonds. At small deformation velocities, microfibrils straighten and slip in relation to each other, whereas during the impact they straighten and break.

There are numerous papers in the literature proving that the apple flesh behaves like a viscoelastic material [37,38,39,40]. This is evidenced by the occurrence of stress relaxation and creep phenomena in these materials. Air spaces can constitute up to 25% of the apple flesh volume [41,42]. There are also liquids in the tissue structures. Hence, fluid flows take place under loading, causing stress relaxation and creep phenomena.

It can be assumed that under the quasi-static loading conditions, there is time for fluids to move through the tissue structures, and their destruction occurs rather when the specified value of the cell wall strain is exceeded. In the case of impact, the fluids do not have enough time to move, which manifests itself in the increased material stiffness. During the impact, the phenomena of stress relaxation and creep occur to the minimum extent. Destruction occurs by exceeding the critical stresses of the cell walls in the fluid-filled tissue structures.

In order to determine the failure mechanisms, it is necessary to determine the parameters that characterize the state of loading in the tested material. These are failure stress and failure strain. Wang et al. [43] obtained a stabilization of the maximum surface pressure with the increasing drop height while investigating the damage susceptibility of litchi fruit during the impact. The constant value of maximum stress for the impact velocities above 1 m⋅s^−1^ was found for pears [44] and potatoes [45]. Similar values of failure strain and toughness for the two ‘Golden Delicious’ and ‘Fuji’ apple cultivars under the quasi-static loading conditions were obtained by Vursavus and Ince [46]. For the ‘Braeburn’ apple cultivar, the tissue toughness under the impact loading conditions was 8.5 kPa [47].

## 5. Conclusions

The study showed that the firmness of the apple flesh influenced the mechanical response under both the quasi-static and impact loading conditions. The elasticity modulus had much higher values under the impact than quasi-static loading conditions. This indicates that the stiffness of the apple tissue was larger during the impact than at the low-speed compression, assuming the constant cross-section of the sample during loading. For the apples with larger firmness, a larger elasticity modulus under the quasi-static loading conditions was found. Under the impact loading conditions, firmness had no effect on the elasticity modulus, having the constant value of about 9 MPa.

Different failure mechanisms were found for the cylindrical apple flesh samples depending on the loading conditions. Under the quasi-static loading conditions, the apples of both cultivars failed at the same strain value of about 0.15. However, during the impact, the apples failed at the constant stress value of 0.4 MPa regardless of the tested cultivar firmness.

The toughness of the apple tissue depended on the firmness and was larger under the quasi-static loading conditions for the apples with greater firmness. However, under the impact loading conditions, toughness was greater for the apples with smaller firmness. This was due to the greater failure stress of the ‘Red Jonaprince’ cultivar under the quasi-static loading conditions, and smaller failure strain under the impact loading conditions compared to the same quantities of the ‘Rubin’ cultivar.

## Figures and Tables

**Figure 1 materials-15-07743-f001:**
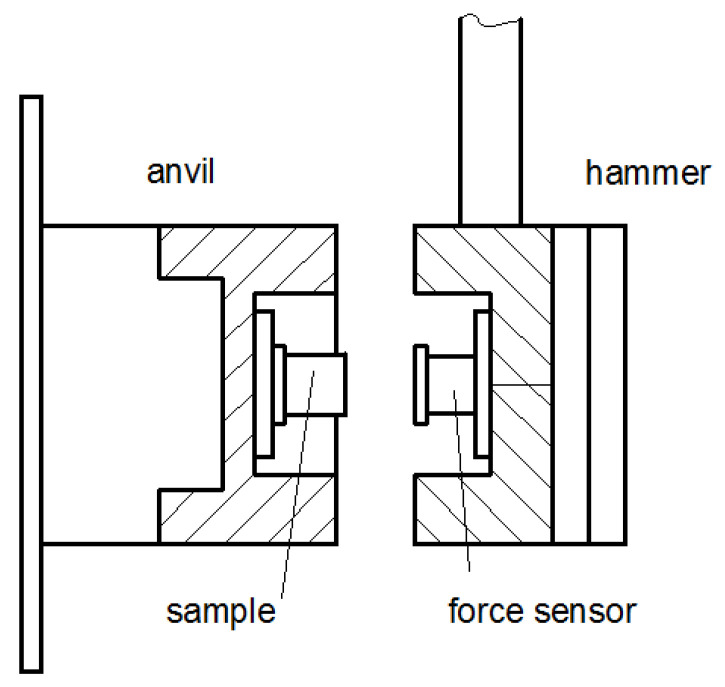
Impact measuring stand.

**Figure 2 materials-15-07743-f002:**
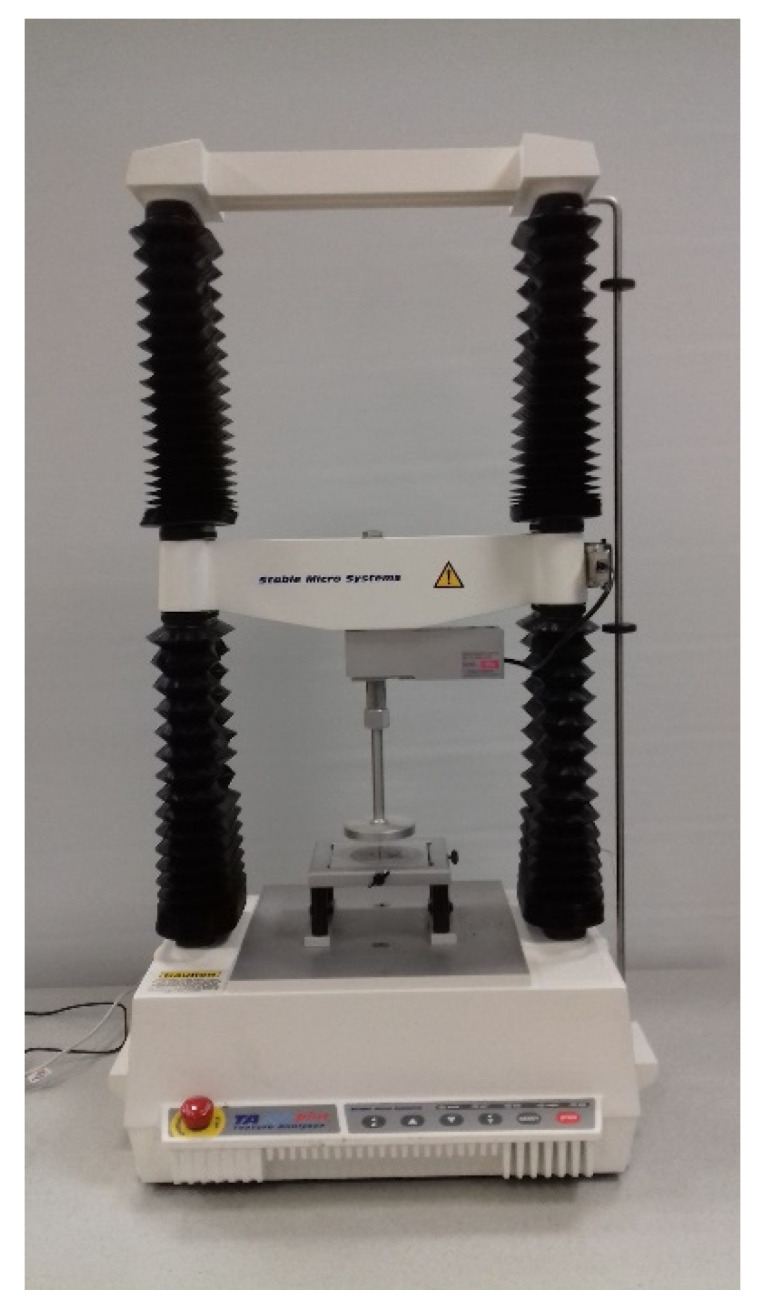
TA.HD plus texture analyser.

**Figure 3 materials-15-07743-f003:**
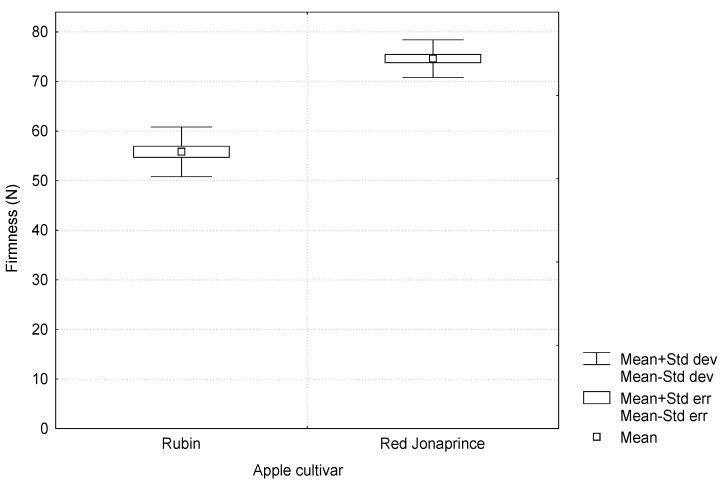
Firmness of the tested apple cultivars.

**Figure 4 materials-15-07743-f004:**
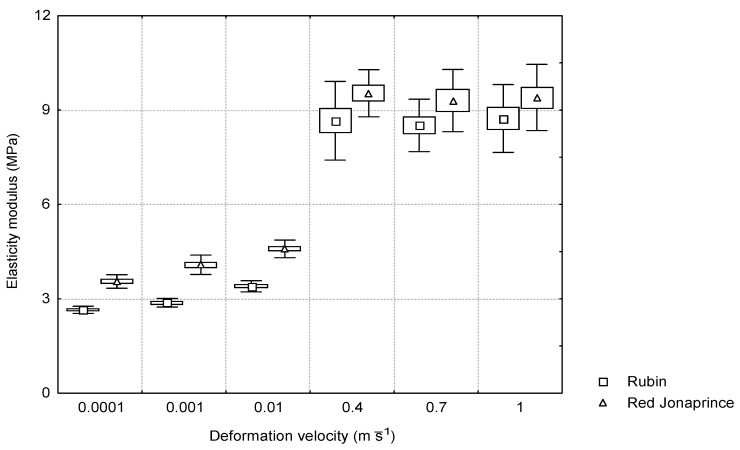
Relationship between the elasticity modulus and the deformation velocity for the two apple cultivars.

**Figure 5 materials-15-07743-f005:**
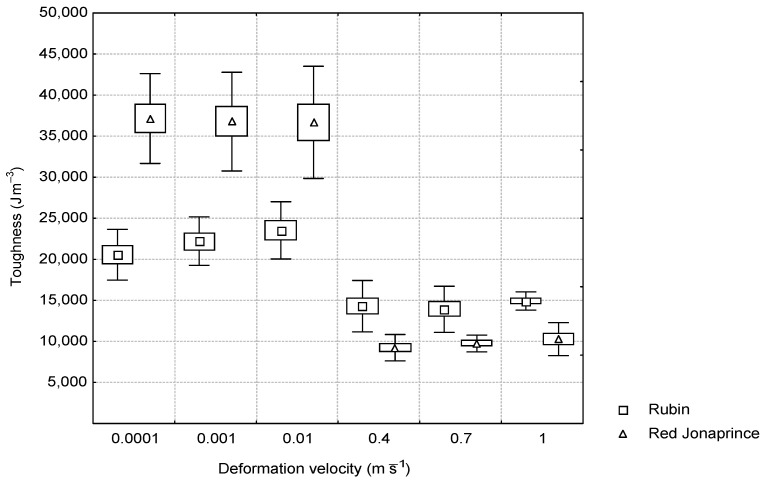
Relationship between the toughness and the deformation velocity for the two apple cultivars.

**Figure 6 materials-15-07743-f006:**
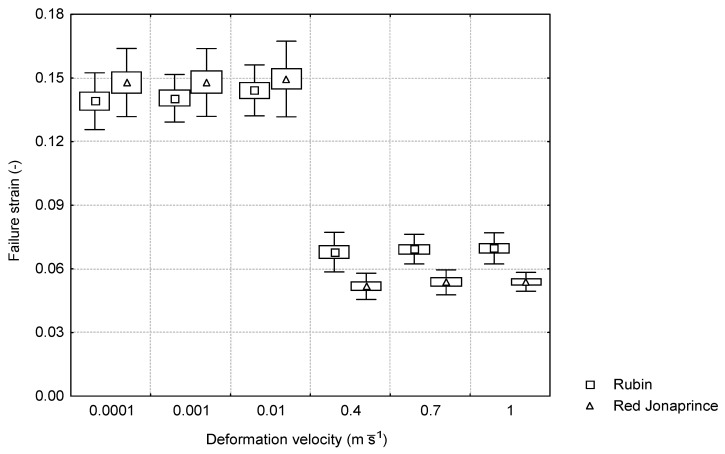
Relationship between the failure strain and the deformation velocity for the two apple cultivars.

**Figure 7 materials-15-07743-f007:**
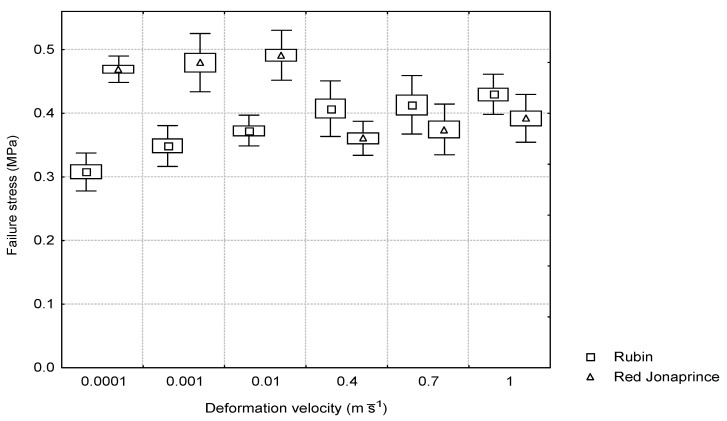
Relationship between the failure stress and the deformation velocity for the two apple cultivars.

## Data Availability

Data are contained within the article.

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
