# Peer review of "Response of Apple Flesh to Compression under the Quasi-Static and Impact Loading Conditions"

_materials, 2022, doi:10.3390/ma15217743_

Round 1

Reviewer 1 Report

I have a few comments I think will enhance the manuscript:

1. the introduction missing the previous studies about the response of apple flesh to compression, meanwhile I found several recent studies on the same point o apple or different crops such as :

- Stropek, Zbigniew & Gołacki, Krzysztof. (2018). Viscoelastic response of apple flesh in a wide range of mechanical loading rates. International Agrophysics. 32. 335-340. 10.1515/intag-2017-0023. 

- Stress Relaxation of Apples at Different Deformation Velocities and Temperatures. Transactions of the ASABE. 10.13031/trans.12993. 2019. Vol 62 (1), pp. 115-121.

2.  section 3.1 firmness measurements; is the first part of the results but when I read it I found it only the determination method, only the last line which was the result!!

3.   all figures without figure caption

4. you didn't cite any previous studies that agree with your results or disagree!

5. conclusion was very poor and short 

Author Response

  1. The additional reference was cited in accordance with the Reviewer's suggestion.

It is denoted [19] in the paper.

  1. The part of chapter 3.1 concerning the firmness determination method was moved to the Materials and Methods chapter. However, the part concerning the test results was supplemented with the text.

"Figure 3 shows the apple flesh firmness for two 'Rubin' and 'Red Jonaprince' cultivars."

  1. In the version sent to the editor, there are captions under each figure. The figure captions are also in the version sent by the editor after the reviews.

  1. The Discussion chapter includes citations of papers [26-30] that refer to similar experiments. In addition, further literature was added. A passage explaining the processes occurring in the plant material under loading was added to complete the discussion.

"There are numerous papers in the literature proving that the apple flesh behaves like a viscoelastic material. This is evidenced by the occurrence of stress relaxation and creep phenomena in these materials. Air spaces can constitute up to 25 % of the apple flesh volume. There are also liquids in the tissue structures. Hence, fluid flows take place under loading causing stress relaxation and creep phenomena.

It can be assumed that under the quasi-static loading conditions there is time for fluids to move through the tissue structures and their destruction occurs rather when the specified value of the cell wall strain is exceeded. In the case of impact, the fluids do not have enough time to move which manifests itself in the increased material stiffness. During the impact the phenomena of stress relaxation and creep occur to the minimum extent. Destruction occurs by exceeding the critical stresses of the cell walls in the fluid-filled tissue structures."

  1. In our opinion, the conclusions should relate strictly to the research results presented in the paper. Moreover, the description of the phenomena interpretation is included in the Discussion chapter.

"There are numerous papers in the literature proving that the apple flesh behaves like a viscoelastic material. This is evidenced by the occurrence of stress relaxation and creep phenomena in these materials. Air spaces can constitute up to 25 % of the apple flesh volume. There are also liquids in the tissue structures. Hence, fluid flows take place under loading causing stress relaxation and creep phenomena.

It can be assumed that under the quasi-static loading conditions there is time for fluids to move through the tissue structures and their destruction occurs rather when the specified value of the cell wall strain is exceeded. In the case of impact, the fluids do not have enough time to move which manifests itself in the increased material stiffness. During the impact the phenomena of stress relaxation and creep occur to the minimum extent. Destruction occurs by exceeding the critical stresses of the cell walls in the fluid-filled tissue structures."

Reviewer 2 Report

In this manuscript, the authors investigate the behavior of apples under mechanical compression at different strain rates, from slow quasi-static loading to fast impact loading. This is an interesting manuscript with useful results. However, there are number of problems that have to be addressed before this manuscript can be published. List of my comments/suggestions is below:

1.      Line 34: Grammatical mistake, I suggest that the authors re-write as: “From tehse curves, the following mechanical parameters can be determined: …”.

2.      Lines 61-65: The authors discuss here different failure types. Generally, type of failure depends on the strain rate, but also on boundary conditions (see my comments #5 below). I suggest that the authors indicate it more clearer and add a couple of references suggested below.

3.      Lines 70-71: It is more correct to use the term strain rate than deformation velocity. The reason is that the strain rate also takes into account the dimensions of the tested specimen.

4.      Lines 94-95: I have some concerns regarding this statement. First of all, the authors say that the specimens were compressed under tension. Is this a typo? If so, please correct it. Secondly, my big concern relates to the fact that the authors indicate that the specimens were loaded uniaxially. However, to ensure uniaxial loading conditions, the specimens have to be significantly longer than their circumferential dimensions (diameter for example), at least 3 times longer than diameter. The authors indicated that their samples have their diameter and length the same. In this case, the stress state is not uniaxial anymore but rather triaxial. I suggest that the authors add and describe this warning for the reader since I imagine it would not be possible to re-do all the tests under proper uniaxial loading. I also imagine that for the practical applications described in the Introduction, there is no need to run uniaxial compression experiments, but rather multi-axial as the same stress state is observed in practice. Therefore, I agree with the choice of dimensions the authors made, but I urge the authors not to use the term “uniaxial” and briefly explain to the reader why the loading that was used is rather multi-axial.

5.      In the description of the compression test setup, the authors have to indicate the type of boundary conditions they used. This is very important and goes along with my previous comment regarding multi-axial stress state during the compression. The reason why this is important is because if we use a rough surface at the loading platen as a boundary, for example, fine grained sandpaper, this increases friction and resists the lateral expansion of the sample, resulting in a confining stress near the loading platen. In addition, placing a rough surface as a boundary condition, can change the mode of failure. For example, if the mode was axial splitting under uniaxial compression, a rough boundary will suppress this mode of failure. On the other hand in order to minimize lateral confinement near the loading platens, a friction-free boundary should be placed between the sample and the loading platens, for example, polyethylene sheets. Generally, in order to obtain uniaxial stress state during loading (this is not the case in current experiments as I already mentioned), friction-free boundary is used. On contrary, when boundary with friction is used, we get a triaxial state of stress near the loading platens and that is why it is required to have a sufficiently long sample to ensure a uniaxial state of stress in its middle part.

This coment also refers to lines 60-65 in the Introduction where the authors discuss the different modes of failures. Not only the strain rate is responsible for the change of failure mode as mentioned in this manuscript in the Introduction, but also boundary conditions can change the mode of failure. I think this is essential to mention briefly in the Introduction and also in the Methods section. The following references have to be added to support this argument (in these papers the authors used both sandpaper as rough boundary condition, first paper, and thin polyethylene sheets, second paper, in order to observe the difference in the behavior; the authors also provided analysis on the observed results):

Renshaw, C. E., Schulson, E. M., Iliescu, D., & Murzda, A. (2020). Increased Fractured Rock Permeability After Percolation Despite Limited Crack Growth. Journal of Geophysical Research: Solid Earth, 125(8), 1–10. https://doi.org/10.1029/2019JB019240

Renshaw, C. E., Murdza, A., & Schulson, E. M. (2021). Experimental Verification of the Isotropic Onset of Percolation in 3D Crack Networks in Polycrystalline Materials With Implications for the Applicability of Percolation Theory to Crustal Rocks. Journal of Geophysical Research: Solid Earth, 126(12), 1–9. https://doi.org/10.1029/2021JB023092

6.      I have some big concerns regarding the Section 3.2 Elasticity modulus. The authors pointed out that the measured modulus of Elasticity was smaller for slow loadings and greater for rapid loadings. However, the authors did not provide any arguments for this other than mentioning somewhere in the text something like this observation is related to the firmness which may be different at different loading rates, which I do not think is the right way to say. The reason is that the elastic moduli can be measured in a static way (using ultrasonic pulse-echo techniques) or in a dynamic way (measuring using tangent of stress-strain curve). The problem with measurements based on the stress-strain curve is that although the curve initially looks linear and we assume that material exhibits linear behavior, this is not correct. Even if it looks linear there is always a creep component (material still has inelastic or plastic deformation in addition to elastic, even if it is small). Therefore, these measuremts tend to be somewhat smaller than measurements received via sending vawes through the sample. This is exactly why the authors observed lower Youngs moduli during slow compression and higher moduli during rapid compression. Indeed, during slow compression we impose more creep deformation to the sample, while during rapid impact there is a lot less plastic deformation introduced, therefore, the measured Youngs modulus is closer to the “real static” value. I think this is very important that the authors provide these arguments in the Section 3.2 and add a few references that support these arguments.

7.      Line 137: Please check if the word “course” is used correctly.

8.      Line 232: There is a grammar mistake in the “there was observed a sudden decrease”. Replace with “a sudden decrease was observed”.

Author Response

  1. The grammatical mistake was corrected.

  1. The chapter 'Compression tests' was supplemented with the additional text. New references were added.

"For both types of tests, the specimens were compressed between two smooth metal plates. The use of smooth surfaces as opposed to rough surfaces did not confine the lateral displacement of the specimen hence the stress state in the specimen was more like unidirectional [1,2]."

  1. Renshaw, C.E.; Schulson, E.M.; Iliescu, D.; Murzda, A. Increased Fractured Rock Permeability After Percolation Despite Limited Crack Growth. J.Geoph. Res.: Solid Earth 2020, 125(8), 1-10.
  2. Renshaw, C.E.; Murdza, A.; Schulson, E.M. Experimental Verification of the Isotropic Onset of Percolation in 3D Crack Networks in Polycrystalline Materials With Implications for the Applicability of Percolation Theory to Crustal Rocks. J. Geoph. Res.: Solid Earth 2021, 126(12), 1-9.

  1. The used sample dimensions are typical of many experiments performed on apples and are derived from their overall dimensions. Furthermore, the velocities expressed in m/s are readily related to the drop height during post-harvest operations. Therefore, we suggest using the quantity 'deformation velocity' in the paper.

  1. The typo 'tension' was corrected.

We agree with the Reviewer's comment that the stress state in the sample is indeed triaxial. The use of much longer specimens is not always possible for an apple. In addition, such a sample has a much larger surface area through which gases and liquids leak from the intercellular spaces and rapid drying occurs. In the experiment we wanted the properties of the sample to be as close as possible to those of the flesh throughout the fruit.

In the paper the term "uniaxial stress state" was removed and the term "compression along the axis" was left.

  1. As suggested by the Reviewer, the compression description was supplemented with the following text:

"For both types of tests, the specimens were compressed between two smooth metal plates. The use of smooth surfaces as opposed to rough surfaces did not confine the lateral displacement of the specimen hence the stress state in the specimen was more like unidirectional."

  1. Many papers prove that the apple flesh has the properties of a viscoelastic material [1-4]. This is evidenced by the stress relaxation and creep phenomena observed in these materials. Air spaces can constitute up to 25 % of the apple flesh volume [5,6]. There are also liquids in the material, so gas and liquid flows can be observed under loading.

Assuming viscoelasticity of the apple flesh in the deformation range to be much below the failure strain or stress, the phenomenon of stress relaxation or creep during the initial deformation can be mathematically taken into account in the stress relaxation or creep test [7,8].

If the tested material was perfectly viscoelastic, then the parameters of the phenomenological viscoelastic model would be the same after taking into account different deformation velocities.

However, the resulting elastic moduli of the viscoelastic model of the real tissue are dependent on the initial deformation velocity. This inadequacy of rheological models (e.g. the generalized Maxwell model) suggests that, even during small deformations, there are micro-damages in the plant material that facilitate fluid flow and cause its properties to be changed during loading. An attempt to interpret these phenomena is included in the paper [9]. We cite these remarks to show the difficulty in interpreting the phenomena occurring in the plant material under loading.

  1. Van Zeebroeck, M.; Dintwa, E.; Tijskens, E.; Deli, V.; Loodts, J.; De Baerdemaeker, J.; Ramon, H. Determining tangential contact force model parameters for viscoelastic materials (apples) using a rheometer. Postharvest Biol. Technol. 2004, 33(2), 111–125
  2. Ahmadi, E.; Barikloo, H.; Kashafi, M. Viscoelastic finite element analysis of the dynamic behavior of apple under impact loading with regard its different layers. Comput. Electron. Agric. 2016, 121, 1-11.
  3. Ji, W.; Qian, Z.; Xu, B.; Chen, G.; Zhao, D. Apple viscoelastic complex model for bruise damage analysis in constant velocity grasping by gripper. Comput. Electron. Agric. 2019, 162, 907-920.
  4. Carillo, S.; Chipot, M.; Valente, V.; Caffarelli, G.V. On weak regularity of the relaxation modulus in viscoelasticity. Commun. Appl. Ind. Math. 2019, 10(1), 78-87.
  5. Baritelle, A.L; Hyde, G.M. Commodity conditioning to reduce impact bruising. Postharvest Biol. Technol. 2001, 21, 331-339.
  6. Pitt, R. E. Viscoelastic properties of fruits and vegetables. In Viscoelastic properties of food, Rao, M.A., Steffe, J.F. (eds.); Elsevier Applied Science; London, 1992, pp. 49-76.
  7. Chen, P. Creep response of a generalized Maxwell model. Int. Agrophys. 1994, 8, 555-558.
  8. Stropek, Z.; Gołacki, K. Viscoelastic response of apple flesh in a wide range of mechanical loading rates. Int. Agrophys. 2018, 32, 335-340.
  9. Stropek, Z.; Gołacki, K. Stress relaxation of apples at different velocities and temperatures. Trans. ASABE 2019, 62(1), 115-121.

Therefore we propose to add the following passage in the Discussion chapter:

"There are numerous papers in the literature proving that the apple flesh behaves like a viscoelastic material. This is evidenced by the occurrence of stress relaxation and creep phenomena in these materials. Air spaces can constitute up to 25 % of the apple flesh volume. There are also liquids in the tissue structures. Hence, fluid flows take place under loading causing stress relaxation and creep phenomena.

It can be assumed that under the quasi-static loading conditions there is time for fluids to move through the tissue structures and their destruction occurs rather when the specified value of the cell wall strain is exceeded. In the case of impact, the fluids do not have enough time to move which manifests itself in the increased material stiffness. During the impact the phenomena of stress relaxation and creep occur to the minimum extent. Destruction occurs by exceeding the critical stresses of the cell walls in the fluid-filled tissue structures."

Round 2

Reviewer 1 Report

All the modification I asked was done، except  for the addition of more recent studies that focused on apple ,  onwhich there are several studies on the same point All the modification I asked was done، except  for the addition more recent studies that focused on apple ,  onwhich there are several studies on the same point 

Author Response

The recent studies on apples were added to the Introduction chapter.

  1. Hussein, Z.; Fawole, O.A.; Opara, U.L. Preharvest factors influencing bruise damage of fresh fruits - a review. Sci. Hort. 2018, 229, 45-58.
  2. Li, Z.; Miao, F.; Andrews, J. Mechanical models of compression and impact on fresh fruits. Compr. Rev. Food Sci. Food Saf. 2017, 16, 1296-1312.
  3. Mahiuddin, M.; Godhani, D.; Feng, L.; Liu, F.; Langrish, T.; Karim, M.A. Application of Caputo fractional rheological model to determine the viscoelastic and mechanical properties of fruit and vegetables. Postharvest Biol. Technol. 2020, 163, 111147.
  4. Zhao, W.; Fang, Y.; Zhang, Q.; Guo, Y.; Gao, G.; Yi, X. Correlation analysis between chemical or texture attributes and stress relaxation properties of 'Fuji' apple. Postharvest Biol. Technol. 2017, 129, 45-51.
  5. Springael, J.; Paternoster, A.; Braet, J. Reducing postharvest losses of apples: Optimal transport routing (while minimizing total costs). Comput. Electron. Agric. 2018, 146, 136-144.
  6. Scheffler, O.C.; Coetzee, C.J.; Opara, U.L. A discrete element model (DEM) for predicting apple damage during handling. Biosyst. Eng. 2018, 172, 29-48.
  7. Stropek, Z.; Gołacki, K. Quantity assessment of plastic deformation energy under impact loading conditions of selected apple cultivars Postharvest Biol. Technol. 2016, 115, 9-17.
  8. Stopa, R.; Szyjewicz, D.; Komarnicki, P.; Kuta, Ł. Determining the resistance to mechanical damage of apples under impact loads. Postharvest Biol. Technol. 2018, 146, 79-89.

Fu, H.; He, L.; Ma, S.; Karkee, M.; Chen, D.; Zhang, Q.; Wang, S. ‘Jazz’ apple impact bruise responses to different cushioning materials. Trans. ASABE 2017, 60(2), 327-336.

Reviewer 2 Report

The authors responded to the reviewer's comments and improved this manuscript. The authors added a sound explanation of why elastic modulus is different during slow and rapid loadings which totally makes sense and I have a full picture now. I suggest this manuscript for publication after a couple of minor comments:

1. The authors should check Figures 3-7. There is something wrong with the formatting and this has to be fixed before the final submission.

2. I think the authors made a typo in the earlier suggested references. One of the names is Murdza, not Murzda (https://agupubs.onlinelibrary.wiley.com/doi/full/10.1029/2021JB023092).

Author Response

  1. It is difficult for us to reply to this remark. The version sent after the review does not contain errors in the figures. This might be due to the conversion of the file from word to pdf.

2. The name in the reference was corrected.